# Different Resistance to DON versus HT2 + T2 Producers in Nordic Oat Varieties

**DOI:** 10.3390/toxins14050313

**Published:** 2022-04-28

**Authors:** Ingerd Skow Hofgaard, Guro Brodal, Marit Almvik, Morten Lillemo, Aina Lundon Russenes, Simon Graham Edwards, Heidi Udnes Aamot

**Affiliations:** 1Division of Biotechnology and Plant Health, Norwegian Institute of Bioeconomy Research (NIBIO), 1431 Ås, Norway; guro.brodal@nibio.no (G.B.); marit.almvik@nibio.no (M.A.); heidi.udnes.aamot@nibio.no (H.U.A.); 2Department of Plant Sciences, Norwegian University of Life Sciences (NMBU), 1432 Ås, Norway; morten.lillemo@nmbu.no; 3Division of Food Production and Society, Norwegian Institute of Bioeconomy Research (NIBIO), 1431 Ås, Norway; aina.lundon@nibio.no; 4Centre for Integrated Pest Management, Harper Adams University, Newport, Shropshire TF 10 8NB, UK; sedwards@harper-adams.ac.uk

**Keywords:** *Fusarium*, DNA, mycotoxins, *Fusarium langsethiae*, *Fusarium graminearum*

## Abstract

Over recent decades, the Norwegian cereal industry has had major practical and financial challenges associated with the occurrence of *Fusarium* head blight (FHB) pathogens and their associated mycotoxins in cereal grains. Deoxynivalenol (DON) is one of the most common *Fusarium*-mycotoxins in Norwegian oats, however T-2 toxin (T2) and HT-2 toxin (HT2) are also commonly detected. The aim of our study was to rank Nordic spring oat varieties and breeding lines by content of the most commonly occurring *Fusarium* mycotoxins (DON and HT2 + T2) as well as by the DNA content of their respective producers. We analyzed the content of mycotoxins and DNA of seven fungal species belonging to the FHB disease complex in grains of Nordic oat varieties and breeding lines harvested from oat field trials located in the main cereal cultivating district in South-East Norway in the years 2011–2020. Oat grains harvested from varieties with a high FHB resistance contained on average half the levels of mycotoxins compared with the most susceptible varieties, which implies that choice of variety may indeed impact on mycotoxin risk. The ranking of oat varieties according to HT2 + T2 levels corresponded with the ranking according to the DNA levels of *Fusarium langsethiae,* but differed from the ranking according to DON and *Fusarium graminearum* DNA. Separate tests are therefore necessary to determine the resistance towards HT2 + T2 and DON producers in oats. This creates practical challenges for the screening of FHB resistance in oats as today’s screening focuses on resistance to *F. graminearum* and DON. We identified oat varieties with generally low levels of both mycotoxins and FHB pathogens which should be preferred to mitigate mycotoxin risk in Norwegian oats.

## 1. Introduction

In most cereal growing regions of the world, wheat, maize, rice and partly barley are the crops that have gained most focus with respect to mycotoxin contamination [1]. In Norway, however, higher contamination of mycotoxins is more often recorded in oats compared to the other cereal species [2,3,4]. This may be a result of growing *Fusarium*-susceptible oat varieties in regions with soil types and climatic conditions optimal for development of mycotoxin producing *Fusarium* species. In 2013, the Norwegian Scientific Committee for Food Safety expressed a concern that the tolerable daily intake for DON may be exceeded in Norwegian infants and children due to consumption of flour and oat flakes in the years with high mycotoxin contamination [5]. *Fusarium graminearum* is regarded as the main DON producer in Norwegian oats [4,6]. Other FHB-related pathogens such as *Fusarium langsethiae, Fusarium avenaceum, Fusarium poae, Fusarium culmorum, Fusarium tricinctum*, *Microdochium majus* and *Microdochium nivale* are also commonly detected in Norwegian oats [4,7,8]. *Fusarium langsethiae* was first described in 2004 [9]. This fungus differs from the other *Fusarium* species as it is a nearly symptomless pathogen in oats [10,11]. *Fusarium langsethiae* is morphologically similar to *F. poae* but has a mycotoxin profile similar to *Fusarium sporotrichioides* [12,13]. *Fusarium langsethiae* is identified as the most important T-2 toxin (T2) and HT-2 toxin (HT2) producer in Norwegian oats [4,6]. Unlike most *Fusarium* species, *M. majus* and *M. nivale* do not produce any known toxic metabolites [14], however, as with many *Fusarium* species, they may cause reduced seedling emergence in cereals, including oats [8,15].

From 2011, payment reductions to Norwegian farmers for oat grain lots with high levels of DON were implemented by the cereal grain buyers. According to the existing legislative maximum permitted level published by the European Commission, oat grain lots with a DON content exceeding 1750 µg/kg shall not be processed for human consumption [16]. Thus, to reduce the risk of elevated DON levels in Norwegian oats, resistance to *F. graminearum* and DON contamination are now routinely tested in varieties and breeding lines [17]. The results of these screenings are taken into consideration before a new variety is officially approved and released to the Norwegian market. However, high levels of HT2 + T2 toxins are also sometimes detected in Norwegian oats [4]. Resistance to *F. langsethiae* and HT2 + T2 contamination is not routinely tested in varieties and breeding lines of oats in Norway.

Surveys of mycotoxin content in oat grains, sampled from farmers’ fields, report no correlation between the content of HT2 + T2 and DON [4,6,18,19,20]. Consequently, high concentrations of T2 + HT2 may be detected in oat grain lots with low DON content, and vice versa. This poor correlation may be a result of HT2 + T2 producing fungi possessing different environmental requirements and/or epidemiology than the DON producers [18,21]. Indeed, the accumulation of DON in oat grain collected from farmers’ fields in Norway was associated with a high average daily precipitation at flowering, whereas HT2 + T2-contamination was associated with a high average daily precipitation during stem elongation and booting [22,23]. Furthermore, cultivation practices may differentially influence the development of DON versus HT2 + T2 [20], and fungicides that reduce the risk of DON in Norwegian oats do not seem to give a similar reduction in HT2 and T2 [24]. HT2 and T2-toxins are considerably more toxic than DON [25]. However, the European Commission has so far only published an indicative limit of 1000 µg HT2 + T2 per kg oat grain [26]. Thus, no routine testing and payment reduction for oat grain lots with high levels of HT2 + T2 have yet been implemented in Norway.

In wheat, a non-species-specific resistance to fungal species within the FHB disease complex has been observed [27,28,29]. Thus, screening for FHB resistance in wheat has often been conducted by using inoculum from solely one of the fungal species that constitutes the FHB disease complex. A similar tendency has been observed in oats in which significant differences in both DON and T-2 levels were observed between varieties, and a correlation was observed in the ranking for both mycotoxins [30]. However, some studies of both wheat and oats indicate a different ranking in the varieties for some pathogens and mycotoxins within the FHB disease complex. In an Italian study of durum wheat, the varieties showed different rates of sensitivity to DON and T2 + HT2 mycotoxins’ accumulation in grains [31]. In a Canadian study, no consistent ranking of oat varieties according to FHB resistance was observed when assessing the percentage *F. graminearum* versus *F. poae* infested kernels [32]. Recently, we observed different rankings for resistance towards DON versus HT2 + T2 producers in three Nordic oat varieties after inoculation in a greenhouse study [33], which indicate that oats may have a species-specific resistance to fungal species within the FHB complex.

The assessment of the Value for Cultivation and Use of varieties of small grain cereals in Norway is carried out by NIBIO in collaboration with the Norwegian Agricultural Advisory Service. A variety can be considered for approval and, if approved, included in the Norwegian official list of varieties after three years of assessment in field trials (https://www.mattilsynet.no/language/english/plants/Plant_varieties/, accessed on 7 December 2020). In 2020, the Norwegian official list comprised 20 oat varieties. Five varieties made up 70% of the total amount of oats seeds sold in Norway in 2020 according to the following ranking: Vinger > Odal > Belinda > Haga > Ringsaker [34]. However, this ranking is constantly changing dependent on the performance and popularity of new varieties entering the market. To reduce the general risk of *Fusarium* and mycotoxins in oats for food and feed, it is important to know the relative ranking of varieties according to resistance to *Fusarium* spp., and thus, contamination of the commonly occurring mycotoxins. Information on resistance to *Fusarium* and mycotoxins enables identification of varieties and breeding lines that can be recommended for cultivation in a Nordic climate due to low risk of mycotoxin contamination. Additionally, it is important to identify varieties that should be withdrawn from the market due to relatively high risk of *Fusarium* spp. and mycotoxin contamination. The aim of our study was to rank Nordic spring oat varieties and breeding lines by content of the most commonly occurring *Fusarium* mycotoxins (DON and HT2 + T2), as well as by the DNA content of their respective producers over a ten-year period. Furthermore, we wanted to assess whether a specific ranking of varieties according to mycotoxin or fungal DNA content was stable across the different mycotoxins, as well as the pathogen species that mainly constitute the FHB disease complex in northern climates.

## 2. Results

### 2.1. Variation in Mycotoxin Levels between Years

The average levels of DON and HT2 + T2 toxins in the harvested oat grains differed between the years and were not following the same trend. This is demonstrated by the annual average mycotoxin levels in the grain from the variety Belinda which was analyzed from all fields each year during the ten-year period (Figure 1). The highest average DON levels in the grains from the variety Belinda were observed in 2011 and 2012, and the levels were also relatively high in 2014, 2016 and 2017. The highest annual average level of HT2 + T2 was observed in 2015, but high average levels were also observed in 2014 and 2019.

### 2.2. Ranking of Oat Varieties According to Mycotoxin Content in Harvested Grain

Five varieties (Belinda, Haga, Odal, Ringsaker and Vinger), were included in all field trials every year of our study. In total, 20 (DON), and 14 (HT2 + T2) naturally infested field trials were included in the statistical analysis. Fields in which the average levels in Belinda were below 100 µg/kg for DON or 25 µg/kg for HT2 + T2 showed limited variation between varieties due to low infection levels and were consequently not included in this analysis. Grain harvested from Ringsaker, Odal and Vinger had DON-levels that were significantly lower and about half of the levels estimated in Belinda (Figure 2). The DON levels of grains harvested from Haga did not significantly differ from the other varieties. Regarding HT2 + T2-toxins, the lowest level was estimated in grain from Vinger (102 µg/kg), significantly lower compared to Odal and Belinda, and less than one third of the HT2 + T2 levels in Odal. Grain harvested from Odal had the highest estimated levels of HT2 + T2 (419 µg/kg), which were significantly different and more than double as high as the levels in Haga, Ringsaker and Vinger. Possible significant differences in DON and HT2 + T2 content of grains harvested from these varieties were calculated by using mixed-effects models in Minitab and Tukey pairwise comparisons. Field site and variety were factors with a significant effect in both models. No correlation was observed between the ranking of these five varieties according to HT2 + T2 versus DON content (R^2^adj = 0%, *p* = 0.55). The output from the mixed-effects model and Tukey pairwise comparisons of DON and HT2 + T2 content in the different varieties was used as input in the regression analysis.

To compare the mycotoxin levels across most varieties, an estimate of the mycotoxin level of a variety or breeding line were only included in the statistical analysis if grains from at least three field sites were analyzed (same criteria for DON and HT2 + T2 levels in Belinda as above). Avetron had the lowest estimated average level of DON (176 µg/kg, Appendix A), significantly lower than the levels in Avanti, Belinda, GN07133, GN07134, GN09078, GN12230, Ivory, Mirella and Symphony. The breeding line GN07133 had the highest estimated average level of DON (1511 µg/kg), which was more than eight times higher than the average level in Avetron. Akseli, Hurdal and Vinger had significantly lower estimated average levels of DON compared to Avanti, Belinda, GN07133 and Ivory. Most breeding lines and varieties had intermediate DON levels which did not significantly differ from each other. According to the mixed-effects model in Minitab, both variety (fixed factor, *p* < 0.001) and field site (random factor, *p* = 0.002) were significantly associated with DON levels (R^2^adj of 74%). When considering the levels of HT2 + T2, Hurum, Vinger and Våler had the lowest estimated levels (81, 102, 106 µg HT2 + T2 per kg grain, respectively), whereas Odal had the highest estimated level of HT2 + T2 (419 µg/kg). Avetron, Canary, GN16174, Haga, Hurum, Hurdal, Ringsaker, Vinger and Våler all had significant lower levels of HT2 + T2 compared to Odal. Belinda had lower levels of HT2 + T2 than Odal, but this was not significant. Twenty-three of the, in total, 39 varieties and lines included in this comparison had intermediate estimated levels of HT2 + T2 which did not significantly differ from other varieties. Both variety (fixed factor, *p* < 0.001) and field site (random factor, *p* = 0.007) were significantly associated with the HT2 + T2 levels according to the mixed-effects model in Minitab (R^2^adj of 77%).

To visualize the ranking of a specific variety according to DON versus HT2 + T2 toxin levels, a scatterplot was drawn (Figure 3). The scatterplot only included varieties for which an average level of both mycotoxins were calculated, which comprised in total 20 of the oat varieties and lines (Appendix A). Akseli, Avetron, Hurdal, Staur, Hurum and Vinger had low levels of both mycotoxins. Belinda, Avanti and GN09078 had high levels of both mycotoxins. Mirella, Symphony, Våler and GN08009 had HT2 + T2 levels below the average level across varieties, but DON levels that were above the average level across varieties. Odal, Dovre, Gimse, Årnes and Haga had DON levels at or below the average level across varieties, but levels of HT2 + T2 exceeding the average level across varieties. The ranking according to the estimated average levels of HT2 + T2 for these 20 varieties and breeding lines was not correlated with the ranking according to the estimated average DON levels (ln-transformed data: R^2^adj = 11%, *p* = 0.09).

### 2.3. Ranking of Oat Varieties According to Fungal DNA Levels in Harvested Grain

To identify possible differences between oat varieties in resistance towards pathogens within the FHB disease complex, we conducted a statistical analysis (mixed-effect model and Tukey comparison) on fungal DNA amounts in grain. The DNA content of a fungal species in oat varieties from a specific field were only included in the analysis if the variety Belinda had a fungal DNA content above 20 pg per µg plant DNA, and an estimate of the fungal DNA level of a variety or breeding line was only included in the statistical analysis if data were obtained from at least three field sites. The levels of *F. culmorum* DNA were highly variable between fields and too low to be included in a separate statistical analysis of possible differences in DNA levels between varieties. The sum of the DNA content of the DON producers (*F. graminearum* and *F. culmorum*) were therefore included in the statistical analysis.

A separate statistical analysis was performed for the seven varieties (Avetron, Belinda, Haga, Odal, Ringsaker, Vinger and Våler) that were included in all the trials from which fungal DNA in the harvested grain was extracted. Significant (*p* < 0.05) differences in average fungal DNA levels between these varieties were observed for the content of *F. graminearum,* the sum of *F. graminearum* and *F. culmorum, F. langsethiae, F. poae, M. majus* and *M. nivale,* whereas a *p* value of 0.06 was obtained for the content of *F. avenaceum* (Table 1). Avetron, Vinger and Våler had low DNA levels of most of the fungal species analyzed, whereas Belinda had moderate to high average DNA levels of all of the fungal species. Odal contained high DNA levels of *F. langsethiae* and relatively low levels of the other fungi. Haga had moderate to high DNA levels, and Ringsaker had moderate to low DNA levels depending on the fungal species analyzed. Field site was a significant factor (*p* < 0.05) in all of the models. We compared the ranking of these seven varieties according to the estimated average DNA levels (ln-transformed values, output from Tukey comparison) of the sum of DNA from the DON producers, *F. culmorum* and *F. graminearum,* versus DNA levels of the various fungal species analyzed. Positive and significant associations were observed between the ranking of these seven varieties according to the estimated average DNA level of *F. culmorum* + *F. graminearum* versus the estimated DNA level of *F. avenaceum* (R^2^adj 47%, *p* = 0.05), and *F. culmorum* + *F. graminearum* versus *F. graminearum* (R^2^adj 97%, *p* < 0.001). The ranking of these varieties according to the estimated average DNA levels of *F. culmorum* + *F. graminearum* was neither associated with the DNA levels of *M. majus* and *M. nivale* (R^2^adj 38–46% and *p* values 0.06–0.08), nor to the DNA levels of *F. langsethiae* and *F. poae* (R^2^adj 0% and *p* values > 0.5). Positive and significant associations were observed between the ranking of varieties according to the estimated average DNA level of *F. avenaceum* versus *M. nivale* (R^2^adj 75% and *p* = 0.007). No other significant associations were detected between the ranking of these seven varieties according to the estimated average DNA levels of the various fungi analyzed. The number of trials included in the statistical analysis differed from 7 to 19 depending on fungal species (Appendix A).

When we compared fungal DNA content in grain from all the varieties, significant (*p* < 0.05) differences in average levels were observed between the varieties for the content of *F. avenaceum*, *F. langsethiae, F. poae,* and *M. nivale,* whereas no significant differences in fungal DNA levels were observed for *M. majus, F. graminearum* and the sum of *F. graminearum* and *F. culmorum* (Appendix A). Avetron had, in general, the lowest estimated DNA levels of most fungal species, and Årnes had moderate to low levels of many of the fungal species. Belinda, Haga and Staur had relatively high estimated levels of DNA for most fungal species, whereas Odal and Ringsaker had high DNA levels of some fungal species and relatively low levels of others. The relative rankings of varieties according to the estimated DNA-levels of the various fungal species are presented in a matrix plot (Appendix A). Positive associations were observed between the ranking of varieties according to the estimated average DNA levels of *F. culmorum* + *F. graminearum* versus *F. graminearum* (R^2^adj 68%, *p* < 0.001, *n* = 18) and *M. majus* (R^2^adj 31%, *p* = 0.03, *n* = 12). Positive associations were also observed between the ranking of varieties according to the estimated average DNA levels of *F. langsethiae* versus *F. poae* (R^2^ adj = 19%, *p* = 0.005, *n* = 36). Furthermore, the ranking of varieties according to the estimated average DNA levels of *M. majus* was significantly correlated with the rankings according to the DNA levels of: *F. graminearum* (R^2^adj 32%, *p*= 0.03, *n* = 12), *F. avenaceum* (R^2^adj 34%, *p* = 0.03, *n* = 12) and *F. poae* (R^2^adj 34%, *p* = 0.03, *n* = 12). For the other combinations, no significant correlations were detected (Appendix A).

### 2.4. Ranking of Oat Varieties According to Mycotoxin versus Fungal DNA Levels

To study whether the ranking of varieties according to mycotoxin levels was related to the ranking according to the DNA levels of the mycotoxin producers, we performed regression analysis on a dataset containing the estimated average values from Tukey pairwise comparisons and mixed-effects models in Minitab (Appendix A). A separate estimate of average fungal DNA levels was performed for seven of the varieties (Avetron, Belinda, Haga, Odal, Ringsaker, Vinger and Våler) which were included in all of the years (2013–2019, Table 1). The ranking according to the estimated DON levels was associated with the overall ranking according to the estimated DNA levels of *F. culmorum* and *F. graminearum* when comparing the seven varieties (R^2^ adj 35%, Figure 4), and when all varieties were included (R^2^ adj 17%, Figure 5B), but the associations were not significant (*p* = 0.1). The varieties Avetron, Vinger, Odal, Ringsaker and Årnes had levels of *F. culmorum* + *F. graminearum* DNA and DON below the median across all varieties, whereas Belinda and GN12230 had levels above the median across all the varieties analyzed.

When we compared the ranking of all varieties according to DON levels with the ranking according to fungal DNA levels of the remaining FHB species (Appendix A), the estimated levels of DON were slightly associated with the DNA levels of *M. nivale* (Figure 5F), however not significant (*p* = 0.1). No clear association was observed between the estimated average levels of DON versus fungal DNA levels of *F. avenaceum* (*n* = 17), *F. langsethiae* (*n* = 22), *F. poae* (*n* = 17) or *M. majus* (*n* = 12) (Figure 5). Odal had DON levels below the median across varieties and moderate to low DNA levels of most fungal species except *F. langsethiae.* Hurum had DON levels below the median value across varieties, but no clear trend regarding the relative DNA levels of the fungal species (Figure 5). Varieties Akseli, Delfin and Staur had DON levels below the median across varieties, but DNA levels above the median values across varieties for most of the fungal species. Most of the varieties with DON levels above the median across varieties had DNA levels above the median values for most fungal species. One exception was breeding line GN12230, which had relatively high DON levels but moderate DNA levels of *F. avenaceum*, *F. poae* and *M. nivale*.

The ranking of oat varieties according to the estimated HT2 + T2 levels was significantly associated with the ranking according to the estimated DNA levels of *F. langsethiae*, both when we performed a separate estimate for the seven above-mentioned varieties (R^2^ adj 92%, *p* < 0.001, Figure 4 right panel), and when we compared the ranking across all the varieties analyzed (R^2^ adj 34%, *p* < 0.001, *n* = 39, Figure 6C). The varieties and breeding lines Avetron, Hurum, Symphony, Vinger, Våler, GN09111, GN14182, GN14070 and GN16174 had levels of HT2 + T2 and *F. langsethiae* DNA below the median across varieties, whereas Belinda, Caddy, Gimse, Haga, Moby, Odal, Poseidon, Ridabu, Årnes, GN09078, GN14209, GN15154, GN16165 and LW06W146-2 (WPB Elyann) had levels above the median across all the varieties analyzed. Akseli, Canary, Hurdal, Mirella, Staur and GN08009 had levels of HT2 + T2 below the median, however DNA levels of *F. langsethiae* above the median across all the varieties analyzed. When we compared the ranking of all varieties according to HT2 + T2 levels with the ranking according to fungal DNA levels of the remaining FHB species (Appendix A), no significant correlation was detected between HT2 + T2 levels and ranking, according to the DNA levels of *F. avenaceum* (*n* = 34), *F. graminearum + F. culmorum* (*n* = 16), *F. poae* (*n* = 34), *M. majus* (*n* = 10) or *M. nivale* (*n* = 23) with R^2^ adjusted values from 0–5% (Figure 6). Some varieties had HT2 + T2 levels and DNA levels of most fungal species below the median across varieties (Avetron, Hurum, Vinger and Våler), but we also observed varieties with relatively high content of HT2 + T2 and DNA levels of most fungal species below the median across varieties (Odal and Årnes) and vice versa (Staur and Canary). Some varieties had both HT2 + T2 levels and DNA levels of most fungal species above the median across varieties (Belinda, Moby and GN16165).

### 2.5. Factors Associated with Mycotoxin Content

To identify factors associated with the mycotoxin content in grains, a mixed-effects model in Minitab was used. Input data were DON or HT2 + T2 levels in grains versus field site, oat variety (Avetron, Belinda, Haga, Odal, Ringsaker, Vinger, Våler) and the DNA content of *F. avenaceum, F. culmorum, F. graminearum, F. langsethiae, F. poae, M. nivale* and *M. majus,* as well as the sum of *F. culmorum* and *F. graminearum* DNA in the harvested grain. The dataset used to identify the factors associated with DON content in grains included data from 11 field trials (Appendix A). The variation in DON was explained by a statistical model in which field site (*p* = 0.02), variety and the sum of *F. culmorum* and *F. graminearum* DNA (*p* < 0.01) was included (R^2^ adj 83%). The dataset used to identify factors associated with HT2 + T2 included 14 trials (Appendix A). The variation in HT2 + T2 was explained by a statistical model (R^2^ adj 83%), comprising field site (*p* = 0.02), variety (*p* = 0.004) and the content of *F. langsethiae* DNA (*p* < 0.001). The DNA content of the remaining fungal species did not have a significant effect in either of these models.

## 3. Discussion

We conducted this study in order to rank Nordic spring oat varieties and breeding lines by their grain content of FHB pathogens and the associated mycotoxins, DON and HT2 + T2, commonly occurring in northern climates, and to assess whether a specific ranking of varieties was stable across these mycotoxins and fungal species. Data from twenty-four naturally infested field trials were used in this study. Five varieties (Belinda, Haga, Odal, Ringsaker and Vinger) were included in all the field trials, and separate statistical analysis were performed to rank these varieties, according to the levels of DON and HT2 + T2 in harvested grain. The DNA content of FHB pathogens in grains was analyzed throughout a seven-year period, and seven varieties were included in all these trials. Thus, separate statistical analysis of fungal DNA levels were performed for these seven varieties. Our dataset was unbalanced since most varieties were not included during the whole period of ten years. However, five varieties were included in almost all trials and can be considered as control varieties. Therefore, we performed a statistical analysis across all varieties to get a rough estimate of the relative ranking of varieties according to mycotoxin and fungal DNA levels. However, less weight should be put on these results due to the large difference in the number of field trials in which a specific variety was included.

The variation in DON content of harvested grains observed in our study was explained by a statistical model including field site, variety and the sum of *F. culmorum* and *F. graminearum* DNA. The DNA content of *F. graminearum* was dominating over *F. culmorum.* The variation in HT2 + T2 was explained by a statistical model including field site, variety and the content of *F. langsethiae* DNA. The DNA content of the remaining fungal species did not have a significant effect on either of these models. Thus, our data support previous findings identifying *F. graminearum* as the major DON producer and *F. langsethiae* as the major HT2 + T2-producer in Norwegian oats [4].

Although many of the varieties had a similar ranking for HT2 + T2 and DON levels, the overall ranking of varieties according to average levels of HT2 + T2 was not correlated with the average levels of DON. The difference in the variety ranking was especially evident for a few varieties such as Odal, which had moderate DON levels but relatively high levels of HT2 + T2, whereas Mirella and Symphony had relatively low levels of HT2 + T2, but high DON levels. The lack of correlation between the overall ranking of varieties according to average levels of HT2 + T2 versus DON in our study are somewhat in contrast to field trials performed in Finland and Germany, where a positive phenotypic correlation in the ranking in oat varieties according to levels of DON versus T2 was observed [30]. In the study by Herrmann et al. [30], the plants were inoculated with *Fusarium* pathogens including *F. langsethiae* and *F. sporotrichioides*. However, it was not clear whether the ranking according to T2 was based on data from the plants infested with one of these pathogens or both. In Norway, *F. langsethiae* is the dominant HT2 + T2 producer in oats [4] and this was also confirmed in our study. The epidemiology of *F. sporotrichioides* probably differs from the one of *F. langsethiae* [35]. As our study hardly had any varieties in common with the study of Herrmann et al. [30], any further comparison appears challenging. In accordance with our study, a similar discrepancy between variety ranking for HT2 + T2 versus DON levels was observed in Norwegian field trials of oats inoculated with *F. langsethiae* and *F. graminearum*, respectively [36]. The ranking for HT2 + T2 versus DON levels in our study was similar to the one obtained in Lillemo et al. [36] for the majority of the varieties. Our study indicate that separate tests are necessary to determine the resistance towards HT2 + T2 and DON producers in oat varieties.

A reason for the discrepancy in the ranking of oat varieties according to average *F. langsethiae*/HT2 + T2 versus *F. graminearum*/DON levels in our study could be that *F. langsethiae* has a different epidemiology than *F. graminearum* [18,21,22,23]. Studies indicate that the time window for the possible influence of weather on *F. langsethiae* infection and mycotoxin contamination in oats differs from the one of *F. graminearum*/DON [22,23]. Oats are susceptible to *F. langsethiae* from heading and onwards [33], but airborne fungal propagules of *F. langsethiae* have been detected later in the growing season, compared to fungal propagules from other species within the FHB complex [37]. The phenotypic characters of the oat varieties included in our experiments could perhaps also explain some of the discrepancy between the variety ranking according to average HT2 + T2 versus DON levels. In a study of mycotoxin content of oat grains, the concentration of HT2 + T2 toxins was up to nine times higher in the small kernel fraction (<2.2 mm) [38], which may indicate that oat varieties with a high proportion of small kernels may be more susceptible to *F. langsethiae* and/or, alternatively, that *F. langsethiae* infection results in a reduced kernel size. Other traits such as plant height, earliness, degree of anther retention, lodging and hull content may also influence the disease resistance towards *Fusarium* spp. [30,39], however such data were not included in our analysis. 

The average yearly levels of DON in the harvested oat grains differed between years and did not follow the same trend as HT2 + T2 toxins. This corresponds with observations on environmental conditions associated with high DON levels which differ from the ones associated with high levels of HT2 + T2 toxins [22,23]. The discrepancy between DON versus HT2 + T2 levels was exemplified by using data from the variety Belinda, from which grains were analyzed for mycotoxin content in about ten field-trials each year throughout the ten-year period. The highest DON levels in Belinda were observed in 2011 and 2012 with levels two–three times the levels in the following years. This is in accordance with data from the Norwegian grain industry, where around 75% of the oat grain lots had DON levels higher than 750 µg/kg in the period from 2010 to 2012, but since then, DON values greater than 750 µg/kg have only been observed in 25% of the grain lots or less [40]. Furthermore, our data are in line with the surveillance program for mycotoxins and fungi in feed materials in Norway [41], where the average DON levels in oats 2009–2012 were more than three times the levels in the years 2013–2020. We surmise that our study on DON content in oat grains is representative for the general mycotoxin risk in Norwegian oats during this time-period. Weather conditions are clearly associated with the DON content in oats [42], and differences in weather conditions between locations within and between years probably had a major influence on DON risk. Additionally, the variety Belinda had more than 50% of the market share in Norway until 2013, but this has been gradually reduced to less than 20% by 2018 [34]. In the same time period, the market share of variety Vinger has increased from less than 1% in 2013, to around 20% from 2017. As Belinda was ranked as one of the most susceptible varieties in our study with average DON levels twice as high as Vinger, we speculate that the increased cultivation of Vinger with a corresponding reduction in cultivation of Belinda may have contributed to a reduction in the general DON levels in Norwegian oats, which underlines the importance of selecting varieties with low mycotoxin risk in order to reduce the general mycotoxin levels.

We observed the highest average levels of HT2 + T2 in Belinda in 2015, with second highest levels in 2014 and 2019. This is in accordance with the surveillance program for mycotoxins and fungi in feed materials in Norway, where the concentrations of HT2 + T2 toxins in Norwegian oats were high in 2014, 2015 and 2019 [41]. In Norway, the temperatures range prior to flowering (https://www.met.no, accessed on 29 March 2022) is optimal for *F. langsethiae* infection [35]. This is exemplified in the growing season of 2014 which was first wet, then dry and warm during oat flowering [43], conditions that seem to favor the development of HT2 + T2 in oats [21,23]. However, we considered that further analysis of a possible association between weather conditions and mycotoxins would be too comprehensive to include and, therefore, should be published separately. Furthermore, the high concentrations of HT2 + T2 in the 2014 harvest coincided with a doubling of the market share for variety Odal from 2013 to 2014 [34]. Odal was introduced to the market in 2012 and regarded as a promising variety due to a high resistance to FHB according to studies on *F. graminearum* and DON [17]. However, this variety was ranked as especially prone to HT2 + T2 in our study. Fortunately, the market share of Odal has now been reduced from 25% in 2018, to 17% in 2020. This example emphasizes the potential risk of unintendedly increasing the risk of some mycotoxins (in this case HT2 + T2 toxins) in oats by selecting and growing oat varieties based on resistance screening for solely one of the fungal species within the FHB complex (in this case *F. graminearum*).

Significant differences in the estimated average DON levels were detected between the oat varieties. Axeli, Avetron, Hurdal, Hurum, Odal, Ringsaker, Staur and Vinger were ranked as the least contaminated varieties, whereas the estimated DON levels were twice as high in varieties such as Avanti, Belinda, Ivory, Mirella and Symphony. For the five varieties that were included in all our field trials, a similar ranking of varieties was observed when we made an estimate on a dataset comprising all varieties compared to a dataset comprising these five varieties, only. However, more significant differences between the varieties were observed in the latter more balanced dataset. The ranking of varieties according to DON content was generally in line with results from a study with *F. graminearum*-inoculated field trials [17]. Mirella was ranked as a variety with a high DON content in our study, in line with the results from Tekle et al. [17], as well as in the Finnish study [39]. Symphony was also ranked as one of the most susceptible varieties in our study and in the study by Tekle et al. [17], which is in contrast to a German study of *F. culmorum*-inoculated oats [30]. However, our study, and that by Herrmann et al. [30], had no additional varieties in common, and a further comparison of the relative ranking of varieties could not be performed. In contrast with our results, no significant differences in the DON content were obtained between Belinda, Odal and Akseli in a Finnish study of *F. graminearum* and *F. culmorum*-inoculated oat field trials [39]. In the study by Hautsalo et al. [39], the DON content of the varieties was about twenty times higher than the concentrations obtained in our study with naturally infested oat trials. The discrepancy between the abovementioned studies highlights the challenges related to the interpretation of results on the relative ranking of varieties according to mycotoxin content as this may differ with environmental conditions including the nature of the *Fusarium* inoculum (various *Fusarium* species, inoculated versus naturally infested field trials, etc.).

The ranking of varieties according to the estimated average DON levels obtained in our study were somewhat associated with the ranking according to the estimated DNA levels of the DON producers. Odal and Ringsaker had low levels of DON as well as low levels of DNA from *F. graminearum* and *F. culmorum*. Yet, some of the varieties with generally low DON levels had relatively high DNA levels of the DON-producers and vice versa. This is exemplified in the variety Avetron, ranked as the variety with the lowest estimated DON levels in our study, but with an average DNA level of *F. graminearum* + *F. culmorum* similar to many of the other varieties and breeding lines. This may be as a result of an underestimated DON level in Avetron, as this variety was not included in the years when DON levels were generally high (2011–2012). In line with those findings, Avetron had moderate DON levels similar to many of the other varieties in a study of *F. graminearum*-inoculated oat varieties [17]. Still, Avetron can be regarded as a promising variety with both DON and DNA levels of the DON producers below the estimated average levels across all varieties. The variety Staur also had DON levels below the estimated average across all varieties. However, this variety may not necessarily be a variety with a low DON risk as it was not tested in the years with high average DON levels. Furthermore, the DNA levels of *F. graminearum* + *F. culmorum* in Staur was above the average across the varieties. Thus, some of the discrepancy in the variety ranking for DON versus DON producers may be as a result of an inappropriate estimate of a variety’s average levels as most of the varieties were not included in all of the years.

For most of the varieties and breeding lines, the estimated average DON levels did not significantly differ from any of the other varieties, which suggest that there is not a large difference in the DON risk between most of the oat varieties and breeding lines included in our study. Some of the varieties had different rankings from field to field which may be a result of a variation within and between fields in environmental-related factors that may impact the DON contamination of oats [44]. As our study was based on data from naturally infested field trials only, the impact of environmental conditions on the DON risk in oats was expected. However, our study confirmed that the DON level of a grain sample is significantly associated with the DNA levels of the respective DON producers, *F. graminearum* and *F. culmorum*. Our study identified Avetron, Odal, Ringsaker and Vinger as moderately resistant towards *F. graminearum* as both the estimated DON levels as well as the estimated DNA levels of DON producing species in these varieties were at, or below, the average level across all varieties in the dataset.

Significant differences in the estimated average levels of HT2 + T2 and *F. langsethiae* DNA were detected between the oat varieties, and the ranking of varieties according to the estimated levels of HT2 + T2 was in accordance with the ranking according to the DNA levels of the HT2 + T2 producer, *F. langsethiae.* Avetron, Hurum, Symphony, Vinger and Våler had low levels of both *F. langsethiae* DNA and HT2 + T2 toxins and could be regarded as relatively resistant towards *F. langsethiae*. In accordance with our study, Symphony had relative low levels of T2 in a Finnish study [30]. The new breeding lines GN14182 and GN16174 had low levels of HT2 + T2 and *F. langsethiae*. Both varieties are high-yielding [34], which indicate these varieties as promising for future oat production. However, preliminary studies indicate elevated DON levels in GN14182 (Graminor, Norway).

No correlation was detected between the ranking of varieties according to HT2 + T2 levels versus the ranking according to the fungal DNA content of any of the fungal species analyzed, other than *F. langsethiae*. In line with those findings, no correlation was detected between the ranking of varieties according to the DNA content of *F. langsethiae* versus the ranking for the DNA content of most of the fungal species (except *F. poae*). This may indicate a species-specific resistance in oats towards selected fungal species (in our study, *F. langsethiae*) within the FHB complex. Likewise, studies of both wheat and oats indicated different variety rankings for some FHB pathogens and mycotoxins [31,32]. In contrast to our study in oats, a non-species specific resistance to fungal species within the FHB complex is observed in wheat [27,28]. In our study, the ranking of varieties according to the DON content was related to the DNA content of *M. nivale*, and the variety ranking for the DNA content of *F. culmorum* + *F. graminearum* was associated with the variety ranking for *F. avenaceum* and for *M. majus*. This may indicate a non-species specific resistance to most of the fungal species within the FHB complex in oats. With regards to *F. poae*, no clear picture was drawn as the ranking of oat varieties according to the DNA content of *F. poae* was correlated with the ranking according to both *F. langsethiae* and to *M. majus*. This may be a result of *F. poae*’s possible role as a secondary invader [29]. Our results are based on naturally infested field trials, which may have contributed to the poor correlation observed between variety ranking according to the DNA levels of the various FHB pathogens. Inoculation experiments should therefore be performed to clarify whether the ranking of oat varieties differs between FHB pathogens.

Some varieties had relatively high levels of both DON and HT2 + T2, as well as relatively high DNA levels of most fungal species. The variety, Belinda, stood out as a highly susceptible variety with high levels of both mycotoxins and FHB pathogens. Further cultivation of this variety should therefore be reduced in order to mitigate mycotoxin risk in oats. Still, breeding lines with similar or even higher levels of DON, HT2 + T2, and DNA of most fungal species were identified in our official field trials. This emphasizes the importance of screening for FHB resistance before a variety is released to the market. Some varieties had relatively low levels of both DON and HT2 + T2, but the DNA levels of most fungal species were above the median values across all varieties. In practice, such varieties may be selected for cultivation due to a reduced risk of mycotoxins. However, due to a relatively high risk of fungal infection, poor germination of seeds from these varieties may still occur after years with high disease pressure of FHB-related fungi.

To enable a reduction in the general risk for elevated levels of mycotoxins in oats for food and feed, it is important to identify oat varieties with high resistance towards mycotoxin-producing fungal species. Only a few of the varieties and lines had levels of DON, HT2 + T2, as well as DNA of most of the fungal species below the median across all of the varieties and lines included in this study. The relatively newly released varieties, Avetron and Vinger, stood out as especially promising varieties with low average levels of DON, HT2 + T2 as well as low DNA levels of all the analyzed FHB pathogens. Some of the new varieties and breeding lines seemed promising with low or moderate levels of FHB pathogens and mycotoxins. However, the relative ranking of varieties observed in this study should not be regarded as absolute. Most of the varieties were not included in all of the field trials performed throughout this ten-year study, and a complete ranking for mycotoxins as well as the DNA of all the fungal species included was only performed for 10 of the, in total, 42 oat varieties and lines. The varieties Avetron, Ringsaker, Vinger and Våler had low levels of mycotoxins as well as low levels of DNA of most of the FHB pathogens analyzed. Thus, we expect that the cultivation of these varieties at the expense of the more susceptible varieties, will mitigate the risk of elevated levels of FHB pathogens and mycotoxins in Norwegian oats. Five varieties (Belinda, Haga, Odal, Ringsaker and Vinger), were included in all of our field trials. Of these five varieties, Vinger had the lowest content of mycotoxins, with levels of DON and HT2 + T2 of about half the levels detected in Belinda. Until recent, Belinda has been the dominating oat variety in Norway, but in 2020 Vinger took over as the most dominating variety [34]. This important shift is expected to mitigate the risk of mycotoxins in Norwegian grown oats.

## 4. Conclusions

To reduce the general risk of mycotoxins in oats for food and feed, it is important to avoid cultivation of FHB-susceptible varieties. Screening for resistance to FHB pathogens in oats is important to identify varieties and breeding lines that can be recommended for cultivation in a Nordic climate, as well as to identify varieties that should be withdrawn from the market due to a high risk of mycotoxin contamination. We have identified the oat varieties with generally low levels of both FHB pathogens and mycotoxins. These varieties might be a good choice for the farmers to mitigate mycotoxin risks in oats. We have shown that the ranking of oat varieties according to content of *F. langsethiae* and HT2 + T2 does not always correspond with the ranking for *F. graminearum* and DON. Separate tests are therefore necessary to determine the risk for HT2 + T2 toxins versus DON contamination in oat varieties. More investigation is needed to clarify whether the ranking of oat varieties according to disease resistance towards DON and HT2 + T2-producers differs from the ranking according to resistance towards other pathogens within the *Fusarium* head blight disease complex.

## 5. Materials and Methods

### 5.1. Field Trials

Our data comprised observations and analysis performed on varieties and lines tested in field trials for the assessment of Value for Cultivation and Use of oats in Norway in the years 2011–2020. These trials are performed at approximately 10 sites each year, predominantly located in the main cereal cultivating district in South-East Norway. Results from the grain quality analysis from these trials are published [34]. The field trials were laid out as block trials with two replicates and the varieties were randomized within each replication. In this paper we present data from, in total, forty-four different oat varieties and breeding lines. However, only the most relevant market varieties were tested each year along with a few new varieties and breeding lines, approximately 20 varieties and lines each year in total. Thus, many of the varieties and breeding lines were only tested for a few years. The varieties were tested without the use of fungicide treatments or growth regulators. The cultivation technique was chosen to simulate the field host’s practice regarding soil management, fertilizer use and weed control. All the varieties within one field site were fertilized equally.

### 5.2. Grain Samples

For each oat variety within a field, approximately one kilo of oat grain harvested from each of the two replicate field plots was mixed and a subsample of 200 g was separated by the use of a riffle divider, milled in a ZM 200 mill fitted with a 1 mm sieve (Retsch). Subsamples of the milled oats were stored at −80 °C upon DNA extraction and at −20 °C upon mycotoxin analysis. Every year we analyzed the content of mycotoxins (mainly DON and HT2 + T2, see method below) in a representative grain sample of the variety Belinda from each field. In fields where the content of mycotoxins in Belinda exceeded 100 µg/kg for DON or 25 µg/kg for HT2 + T2, grain samples of most varieties and breeding lines were analyzed for the content of mycotoxins and *Fusarium* DNA (selected years).

### 5.3. Mycotoxin Analysis of Harvested Grains

In 2011–2017, the content of DON and HT2 + T2 in harvested oat grain was analyzed by ELISA using the kits AgraQuant^®^ Deoxynivalenol Assay and AgraQuant^®^ T2/HT2 Assay (both produced by Romer Labs^®^, Tulln, Austria), according to the procedure described in [22,23]. In short, 25 mL of extraction liquid (distilled water in case of DON, and 70% methanol in case of HT2 + T2) was added to 5 g of ground sample. The mixture was vigorously shaken for 3 min then centrifuged for 1 min at 1811× *g*. For DON analysis, the supernatant was diluted 1 + 3 with distilled water, and further analyzed, according to the manufacturer’s protocol. For HT2 + T2 analysis, the supernatant was diluted 1 + 9 with distilled water. To obtain a limit of detection (LOD) of 120 μg/kg and a limit of quantification (LOQ) of 250 μg/kg for the HT2 + T2 kit, an additional dilution of 1 + 1 with 7% methanol was included, that was further analyzed, according to the manufacturer’s protocol.

In the years 2018–2020, the content of DON and HT2 + T2 toxins was analyzed by the use of liquid chromatography coupled to high resolution mass spectrometry [38]. The sample preparation was completed according to the procedure published by Klötzel and Lauber [45] except that only 5 g aliquot of each sample was extracted with 20 mL mixture of acetonitrile and water (80:20 *v*/*v*). The extracts were cleaned up by passing through a Bond Elut Mycotoxins column (500 mg bed mass, Agilent Technologies) and a 2 mL aliquot evaporated to dryness and redissolved in 1 mL 25% acetonitrile in water. Samples were filtered (VWR PTFE 0.2 µm) to vial and analyzed the same day with LC-HRMS in the targeted SIM data-dependent MS2 mode. DON was detected as acetate-adduct [M + CH3COO]^−^ and HT2 and T2 as ammonium adducts [M + NH4]^+^ in the same run, using 5 mM ammonium acetate in water and in methanol, respectively, as mobile phases. The mycotoxins were identified based on retention time (RT) match to reference standard (± 0.1 min), precursor ion m/z mass match within 5 ppm accuracy and the presence of at least one targeted product ion within 5 ppm accuracy and produced by fragmentation of the precursor ion. The method had a LOQ of 1 µg/kg. Control samples were prepared with every batch and the recovery of HT2, T2 and DON over the period was 97–103%, using a certified oat reference material from BAM (ERM-BC720; 81 µg/kg T2 and HT2, containing also a non-certified level of 55 µg DON/kg). The method was shown to report the correct levels of HT2 (assigned values 61 and 185 µg/kg, z-scores ≤ 0.35), T2 (assigned values 24 and 133 µg/kg, z-scores ≤ 0.34) and DON (assigned values 4174 and 4268 µg/kg, z-scores = 0.07) in oat meal in a proficiency test in 2019 [46]. Z-scores ≤ 2.0 are generally regarded as satisfactory, and our method achieved z-scores ≤ 0.35.

### 5.4. Fungal DNA Content of Harvested Grains

In the years 2013–2019, the fungal DNA content in grains was analyzed by quantitative PCR (qPCR) for the following fungal species: *Fusarium avenaceum* (Fa), *Fusarium culmorum* (Fc), *Fusarium graminearum* (Fg), *Fusarium langsethiae* (Fl), *Fusarium poae* (Fp), *Microdochium nivale* (Mn) and *Microdochium majus* (Mm). In addition, the host plant DNA was quantified in each sample. 

Total genomic DNA was extracted from 150 mg of flour using a FastDNA SPIN Kit for Soil (MP Biomedicals, Solon OH, USA), following the manufacturer’s directions. DNA was eluted in a volume of 100 µL. The DNA was analyzed with qPCR using the probes and primers shown in Table 2. The *M. nivale* assay was a SYBR Green assay, all others were probe assays. The qPCR reactions were performed according to Aamot et al. [47], in a total volume of 25 µL that consisted of 4 µL genomic DNA from wheat samples (diluted 1 + 9 with PCR grade water) or DNA from pure cultures (standards). Assays for Fg, Fp and Mm were probe assays that were run in singleton reactions, and included 300 nM of each primer, 100 nM probe and Sso Advanced™ Universal Probes Supermix (Bio-Rad, Hercules, CA, USA). The DNA from Fa and Fc was analyzed in duplex reactions that consisted of 300 nM pf each forward- and 100 nM reverse-primer, and 100 nM of each probe. The DNA from host plant and Fl was also performed in duplex reactions, these contained 300 nM of each Fl primer, 75 nM of each plant primer, 100 nM of each probe. The iQ™ Multiplex Powermix (Bio-Rad) was used for the duplex reactions. The SYBR reactions (Mn) consisted of 300 nM of each primer and Sso Advanced™ Universal SYBR^®^ Green Supermix (Bio-Rad). The reactions were run in a C1000 Touch Thermal Cycler combined with a CFX96TM Real-Time System (Bio-Rad, Hercules, CA, USA) using the following parameters: 95 °C for 3 min, followed by 45 cycles of 95 °C for 10 s and 60 °C for 30 s.

Genomic DNA from pure cultures of the different fungi was extracted, as described in [48]. For quantification of DNA from the different fungi, five serial dilutions in the range 1–4000 pg of DNA from pure cultures of the respective species were used. For the quantification of host plant DNA, the serial dilution contained plant DNA in the range 0.08–32 ng. The amount of fungal DNA was normalized against the amount of plant DNA and presented as pg fungal DNA per ng plant DNA (pg/μg).

### 5.5. Statistical Analysis

Grain harvested from the different field trials was analyzed for mycotoxins (DON and/or HT2 + T2) in the years 2011–2020, and for the DNA content of selected fungal species in the years 2013–2019 (Appendix A). Possible differences in the average content of mycotoxins (DON and HT2 + T2) or fungal DNA between the varieties and breeding lines included in this study was estimated by using the mixed-effects model and Tukey pairwise comparisons with 95% confidence in Minitab (MINITAB^®^ 19.2). The field site was used as a random factor and variety as a fixed factor in the statistical models. Ln-transformed data of mycotoxin content or fungal DNA content + 1 were used as input in all the analyses in order to achieve residual plots with no critical deviations from the assumptions of the response variables being normally distributed with homogeneous variance.

The dataset used in the statistical analysis of possible differences in mycotoxin content between varieties included only the field sites in which the average mycotoxin levels in Belinda were above 100 µg/kg for DON and/or above 25 µg/kg for HT2 + T2. In the analysis of possible significant differences in the fungal DNA content between varieties or breeding lines, the data regarding the DNA content of a fungal species in oat varieties from a specific field were only included in the analysis if Belinda had a DNA content above 20 pg per µg plant DNA, to ensure a certain infection level in the included grain samples. An estimate of the average mycotoxin or fungal DNA levels of a variety or breeding line were only included in the statistical analysis if grains from at least three field sites were analyzed.

The varieties Belinda, Haga, Odal, Ringsaker and Vinger were grown in all the field trials included in this study. Thus, a mixed-effects model was run separately to identify possible significant differences in mycotoxin content (DON and HT2 + T2) of oat grains at harvest between these varieties. The varieties Avetron, Belinda, Haga, Odal, Ringsaker, Vinger and Våler were included in all the field trials from which fungal DNA were analyzed. Thus, a mixed-effects model (MINITAB^®^ 19.2) was run separately to identify possible significant differences in Fungal DNA content of oat grains at harvest between these varieties. A variety’s estimated average mycotoxin or fungal DNA level, shown in the Figures and Tables, was calculated by back-transforming the ln output from the statistical models.

To study the associations between the ranking of oat varieties according to the content of DON, HT2 + T2 and the various fungal species, we performed regression analysis (MINITAB^®^ 19.2). Our dataset was unbalanced since most varieties were not included during the whole period of ten years. Thus, instead of using average values calculated across field trials, the estimated average values of DON, HT2 + T2 and fungal DNA content from the output of the mixed-effects model in Minitab were used as input in the regression analysis.

To test whether the mycotoxin content in grains at harvest was associated with the DNA content of selected fungal species, in addition to field site and variety, a mixed-effects model was used. The varieties Avetron, Belinda, Haga, Odal, Ringsaker, Vinger, and Våler were included in this analysis since they were included in all the field trials from which both fungal DNA content and mycotoxins were analyzed, in the years 2013–2019 (Appendix A). The contents of mycotoxins (DON+ and/or HT2 + T2) were used as response variables, and the DNA content of each of the various fungal species as well as the sum of *F. culmorum* and *F. graminearum* DNA in harvested grains were used as covariables in the statistical analysis. Ln-transformed data of the mycotoxin content + 1 or fungal DNA content + 1 from all the analyzed fields were used in this analysis. Covariables which had no significant effect (*p* > 0.05) were excluded from the model. Tukey pairwise comparisons and 95% confidence were used to rank the varieties according to the mycotoxin content (DON and HT2 + T2) and fungal DNA content.

## Figures and Tables

**Figure 1 toxins-14-00313-f001:**
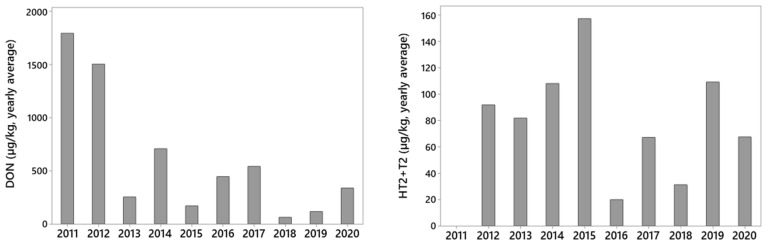
The annual average levels of DON (**left** figure) and HT2 + T2 toxins (**right** figure) in oat grains from the variety Belinda harvested from 10–11 naturally infested field trials in Norway over a ten-year period (2011–2020). In 2011, the content of HT2 + T2 was not analyzed.

**Figure 2 toxins-14-00313-f002:**
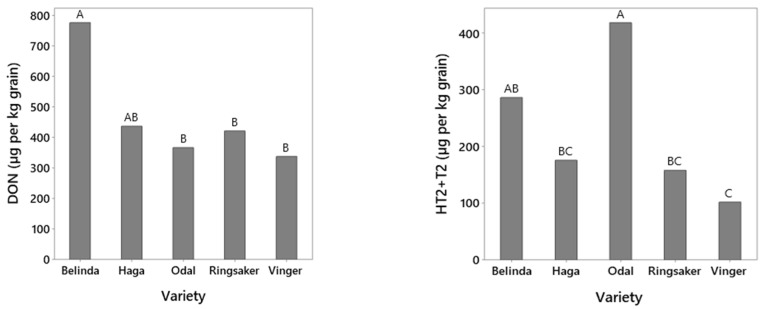
The estimated average mycotoxin level (µg/kg grain) in grains from five oat varieties grown in the years 2011–2020. DON = deoxynivalenol; HT2 + T2 = HT-2 and T-2 toxins. The grain samples analyzed for mycotoxins were harvested from, in total, 20 (DON), and 14 (HT2 + T2) naturally infested field trials in Norway (figure **left** and **right**, respectively). Ln-transformed data on mycotoxin content in harvested grain were used as input in the statistical analysis by mixed-effects model and Tukey pairwise comparisons in Minitab. The values shown in this figure are back-transformed from the model output (ln estimated mycotoxin levels). Varieties with the same letter are not significantly different (Tukey, *p* = 0.05).

**Figure 3 toxins-14-00313-f003:**
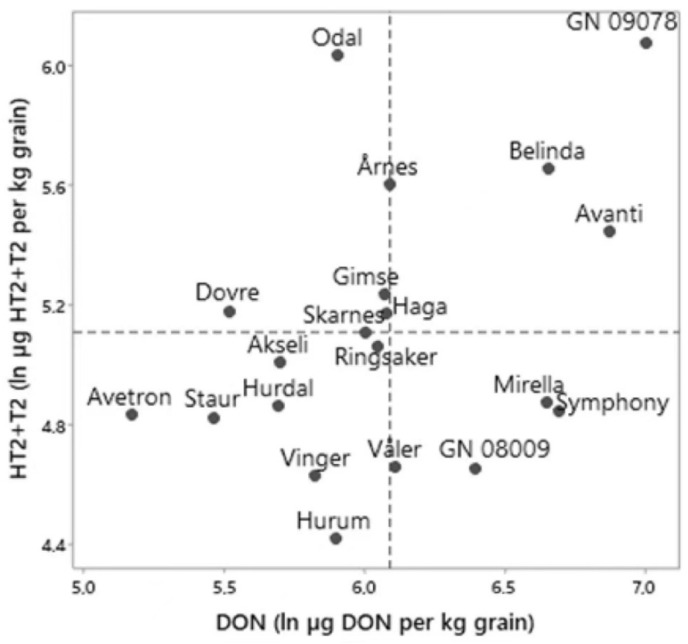
The estimated average level of HT2 + T2 versus DON (shown in ln µg mycotoxin per kg grain) for 20 varieties and breeding lines of oats grown in naturally infested field trials in Norway 2011–2020. The average mycotoxin levels shown for each variety is the output after data analysis by mixed-effects models and Tukey pairwise comparisons in Minitab. Dotted lines indicate the average mycotoxin levels across varieties. For each variety, the number of fields from which the mycotoxin level in harvested grain were analyzed is indicated in Appendix A.

**Figure 4 toxins-14-00313-f004:**
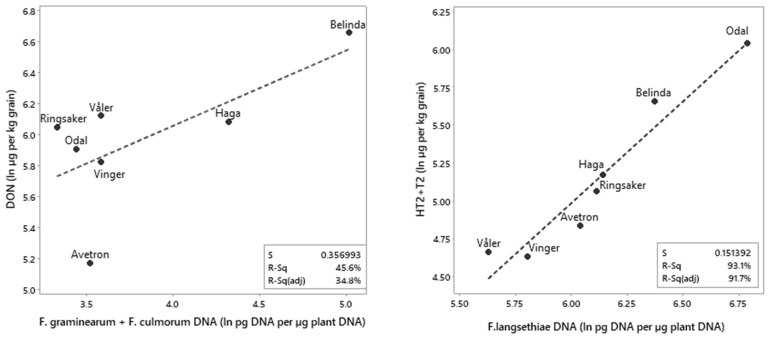
Ranking of seven oat varieties according to mycotoxin levels in grain (ln µg mycotoxins per kg grain) harvested from naturally infested field trials in Norway versus DNA levels of the fungal species producing these mycotoxins (ln pg fungal DNA per µg plant DNA). The estimated average levels of deoxynivalenol (DON) versus the estimated average fungal DNA levels of *F. culmorum* + *F. graminearum* (**left** figure). The estimated average levels of HT2 + T2 versus the estimated average fungal DNA levels of *F. langsethiae* (**right** figure). The estimated average mycotoxin or fungal DNA level for each variety included in this analysis is the output from Tukey pairwise comparisons after data analysis by mixed-effects models in Minitab.

**Figure 5 toxins-14-00313-f005:**
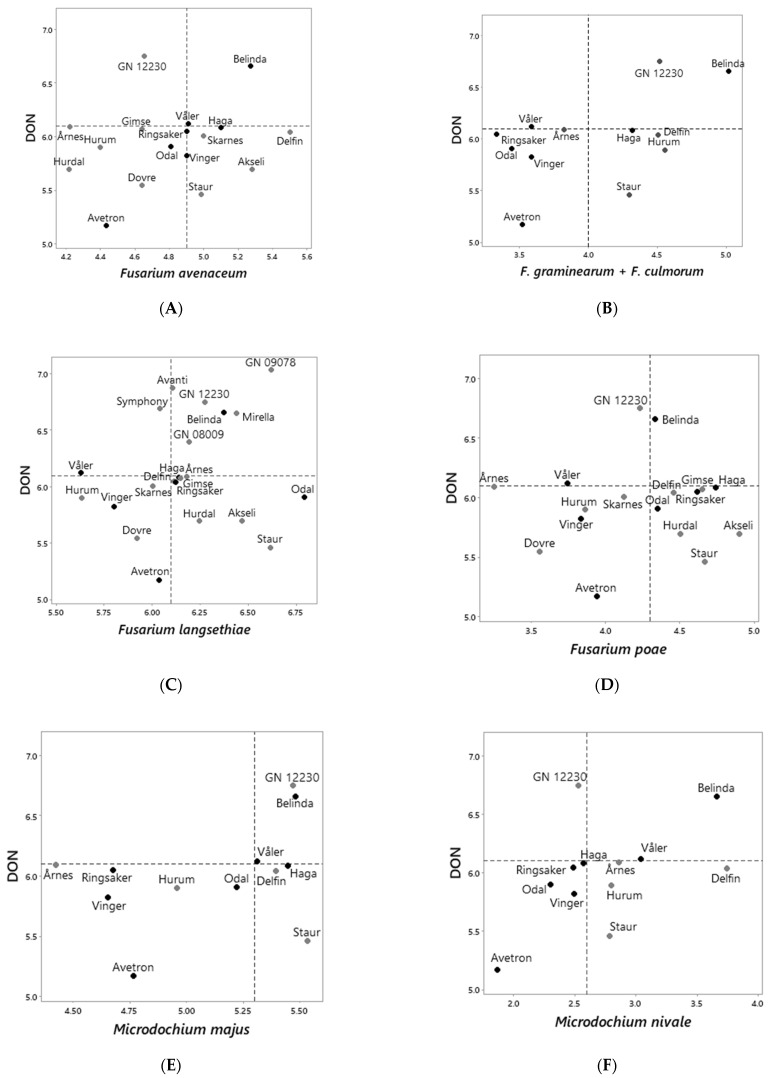
Ranking of oat varieties and breeding lines according to the estimated average levels of DON (ln µg DON per kg grain) versus fungal DNA levels in harvested grains (ln pg fungal DNA per µg plant DNA) estimated for: *Fusarium avenaceum* (**A**); *F. graminearum + F. culmorum* (**B**); *F. langsethiae* (**C**); *F. poae* (**D**); *Microdochium majus* (**E**)*,* and *M. nivale* (**F**). Fungal DNA and mycotoxin content was analyzed in grains harvested from plants grown in naturally infested field trials in Norway, years 2011–2020. The estimated average mycotoxin or fungal DNA level for each variety included in this analysis is the output from Tukey pairwise comparisons after statistical analysis by mixed-effects models in Minitab. Black symbols indicate varieties that were included in all the field trials from which the fungal DNA content of harvested oat grains was analyzed, years 2013–2019. Stippled lines indicate the median values across all the varieties analyzed.

**Figure 6 toxins-14-00313-f006:**
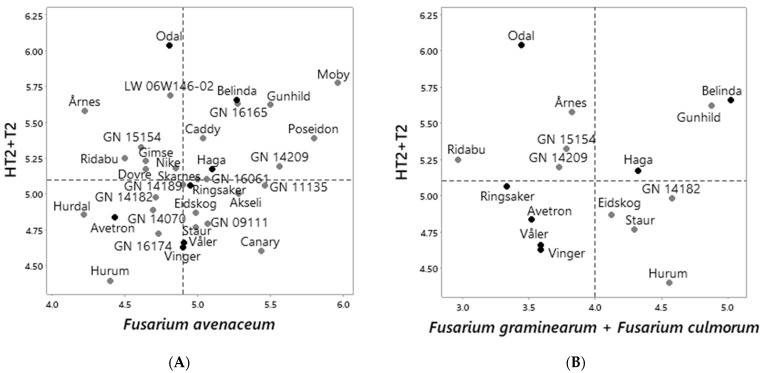
Ranking of oat varieties and breeding lines according to the estimated average levels of HT2 + T2 toxins (ln µg HT2 + T2 per kg grain) versus levels of fungal DNA in harvested grains (ln pg fungal DNA per µg plant DNA) estimated for: *Fusarium avenaceum* (**A**); *F. graminearum + F. culmorum* (**B**); *F. langsethiae* (**C**); *F. poae* (**D**); *Microdochium majus* (**E**) and *M. nivale* (**F**). Fungal DNA and mycotoxin content was analyzed in grains harvested from plants grown in naturally infested field trials in Norway, years 2011–2020. The estimated average mycotoxin or fungal DNA level for each variety included in this analysis was the output from Tukey pairwise comparisons after statistical analysis by mixed-effects models in Minitab. Black symbols indicate varieties that were included in all the field trials from which the fungal DNA content of harvested oat grains was analyzed, years 2013–2019. Stippled lines indicate the median values across all the varieties analyzed.

**Table 1 toxins-14-00313-t001:** The estimated average fungal DNA content (pg fungal DNA per µg plant DNA) in grain from seven oat varieties grown in naturally infested field trials, years 2013–2019.

Oat Variety ^1^	FaDNA (18) ^2,3^		FcFgDNA (10)		FgDNA (10)		FlDNA (19)		FpDNA (16)		MmDNA (8)		MnDNA (7)	
Avetron	84	B	34	AB	33	AB	419	BCD	51	AB	118	A	21	C
Belinda	194	A	151	A	123	A	586	AB	76	AB	240	A	420	A
Haga	164	AB	75	AB	57	AB	464	BC	115	A	233	A	77	BC
Odal	123	AB	31	B	31	AB	889	A	78	AB	186	A	35	BC
Ringsaker	135	AB	28	B	23	B	453	BCD	101	AB	108	A	63	BC
Vinger	134	AB	36	AB	35	AB	332	CD	46	AB	105	A	80	BC
Våler	135	AB	36	AB	36	AB	278	D	42	B	203	A	115	AB
P ^4^	0.06		0.01		0.03		<0.001		0.01		0.02		<0.001	

^1^: Seven varieties of oats harvested from field trials in years 2013–2019, were included; ^2^: The estimated average fungal DNA content (pg fungal DNA per µg plant DNA) of the following fungal species Fa = *Fusarium avenaceum*; Fc = *Fusarium culmorum*; Fg = *Fusarium graminearum*; FcFg = sum of DNA from Fc + Fg; Fl = *Fusarium langsethiae*; Fp = *Fusarium poae*; Mm = *Microdochium majus* and Mn = *Microdochium nivale*. The number of field trials from which data are included is indicated in parenthesis. The estimated values of fungal DNA content are back-transformed values from the output of the Tukey pairwise comparisons. To ascertain a minimum inoculum level of each fungal species, only data from fields in which the fungal DNA content exceeded 20 pg per µg plant DNA in Belinda were included in the statistical analysis. The fungal DNA content was quantified by qPCR. ^3^: Means that do not share a letter are significantly different according to Tukey pairwise comparisons and 95% Confidence. The DNA values were ln-transformed prior to statistical analysis using mixed-effects model in Minitab in which variety was used as fixed factor and field as random factor. ^4^: The statistical significance of variety as a fixed factor in the statistical model.

**Table 2 toxins-14-00313-t002:** Primers and probes used in quantitative PCR for the detection of DNA from different *Fusarium* or *Microdochium* species, or cereal DNA, in oat samples.

Target Species ^1^	Ref. ^2^	Primers, Probe	Primer/Probe Sequences
*F. avenaceum*, *F. arthrosporoides*	[49]	TMAVf	AGATCGGACAATGGTGCATTATAA
		TMAVr	GGCCCTACTATTTACTCTTGCTTTTG
		TMAVp	Cyanine5-CTCCTGAGAGGTCCCAGAGATGAACATAACTTC-BHQ3
*F. culmorum*	[50]	culmorum MGB-F	TCACCCAAGACGGGAATGA
		culmorum MGB-R	GAACGCTGCCCTCAAGCTT
		culmorum MGB pr	6FAM-CACTTGGATATATTTCC-MGBNFQ
*F. graminearum*	[50]	graminearum MGB-F	GGCGCTTCTCGTGAACACA
		graminearum MGB-R	TGGCTAAACAGCACGAATGC
		graminearum MGB pr	6FAM-AGATATGTCTCTTCAAGTCT-MGBNFQ
*F. langsethiae*	[4,51]	Flan forw	GTTGGCGTCTCACTTATTATT C
		Flan rev	TGACATTGTTCAGATAGTAGTCC
		Flan probe	6FAM-CACACC[+C]ATA[+C]CTA[+C]GTGTAA-TAMRA
*F. poae*	[52]	TMpoaef	GCTGAGGGTAAGCCGTCCTT
		TMpoaer	TCTGTCCCCCCTACCAAGCT
		TMpoae probe	TexasRed-ATTTCCCCAACTTC GACTCTCCGAGGA-BHQ2
*M. majus*	[50]	nivale 2-F	CGCCAAGGACTCCTCCAGTAG
		nivale 2-R	GCCGACGAATGGATATTAAGAACT
		nivale 2 probe	6FAM-TCCCGCCTTCACGGTGGAAAGC-TAMRA
*M. nivale* (SYBR)	[53]	Mniv1f	TTGGCTTGCACAAACAATACTTTTT
		Mniv1r	AGCACAACAGGCGTGGATAAG
Cereals	[10]	Cox554f	GGTTGTTGCCACCAAGTCTCTT
		Cox554r	TGCCGCTGCCAACTTC
		Cox554p	VIC-CTCCTATTAAGCTCAGCCTT-MGBNFQ

^1^: Target species that were amplified using SYBR technology are marked (SYBR), all other targets were amplified using a dyed probe; ^2^: Publications in which the different primers and probes are presented.

## Data Availability

All data used in and created by this study are included in this publication as tables, figures, and Appendix A.

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
