# Peer review of "Different Resistance to DON versus HT2 + T2 Producers in Nordic Oat Varieties"

_toxins, 2022, doi:10.3390/toxins14050313_

Round 1

Reviewer 1 Report

The topics discussed in the article have a practical aspect, important not only for oat producers, but above all for all consumers, because the quality of food affects human health.

 The article 'Different resistance to DON versus HT2 + T2 producers in Nordic oat varieties' concerned the analysis of the most common Fusarium mycotoxins (DON and HT2 + T2) in oat varieties grown in Norway. The research was carried out on the available material from 10 years of field experience. The analysis of heterogeneous data, i.e. grain naturally infected and of different varieties, is a research material that is difficult to analyze, and the authors dealt with it well. However, the interpretation of results and conclusions requires great care, as many factors may influence the growth and development of fusariums.

The authors have developed the chapters in detail and comprehensively: 1. Introduction, 3. Discussion and 6. Materials and Methods.

However, some chapters require more details: 2. Results 

 - In the results chapter 2.1. Variation in mycotoxin levels between years - there is no analysis of weather data in the context of the assessment of mycotoxin levels in oat grain. It is important due to the large variation in the level of toxins in oat grain in the studied years (Fig. 1). The authors described in the introduction (verses 61-65) that environmental conditions can determine the accumulation of mycotoxins in the grain. Therefore, it is worth checking which factor was more important for the development of FHB and the accumulation of mycotoxins in the grain, was the specificity of the site (field) or weather conditions a significant factor?

- In chapter 2.3. Ranking of oat varieties according to fungal DNA levels in harvested grain.

It seems unreasonable to look for a relationship between the DNA level of F. culmorum + F. graminearum compared to F. graminearum (R2adj 68%, P <0.001, n = 18). (verse 254-256)

2.5. Factors associated with mycotoxin content - This chapter requires more detailed analysis 

Author Response

Thank you for the constructive and positive feedback you gave on our manuscript. Based on your feedback, we have now corrected the manuscript.

Mycotoxins in grain versus weather factors:

We agree that it is important to assess to what extent the weather conditions at the different field sites are associated with the development of FHB and the accumulation of mycotoxins in the grain. However, as we find this assessment to be quite comprehensive, we have decided not to include it in this manuscript. We are planning to publish another manuscript in which the quality of oat grain harvested from the various locations will be studied in relation to weather conditions, oat variety etc

We have therefore included the following explanation in the discussion, page 13.: “However, we considered that further analysis of possible association between weather conditions and mycotoxins would be too comprehensive to include and therefore, should be published separately.”

The relationship between the DNA level of F. culmorum + F. graminearum compared to F. graminearum:

The reason why we have studied the relationship between the DNA level of F. culmorum + F. graminearum compared to F. graminearum, is to visualize that the inclusion of F. culmorum DNA, despite low DNA levels, may have an influence on the ranking of varieties according to the DNA content of DON-producers.

Reviewer 2 Report

I have carefully read the article “Different resistance to DON versus HT2+T2 producers in Nordic oat varieties”. High economic losses of cereal industry, but also health risk coming from contaminated cereal consumption due to FHB pathogens and associated mycotoxins, indicate the importance of identifying cereal crop varieties to be recommended for cultivation in a particular climate, along with those that should be withdrawn from the market due to a high risk of mycotoxin contamination. The objective of this study was to rank Nordic spring oat varieties and breeding lines by content of the most frequent Fusarium mycotoxins DON and HT2+T2, as well as by the DNA content of their respective producers over a ten-year period. The study revealed oat varieties with generally low levels of both FHB pathogens and mycotoxins, but also showed that ranking of oat varieties according to content of F. langsethiae and HT2+T2 not always corresponds with the ranking for F. graminearum and DON, emphasizing the potential risk of unintendedly increasing the risk of certain mycotoxins in oats by selecting and growing oat varieties based on resistance screening for solely one of the fungal species within the FHB complex. Additional research is needed to clarify whether the ranking of oat varieties according to disease resistance towards DON and HT2+T2 producers differs from the ranking according to resistance towards other pathogens within the FHB disease complex.

I found the topic of the article interesting, and the manuscript itself well written and organized. The cited references are mainly current, results are clearly presented and discussed, containing publishable data. English language and style are generally satisfying and the manuscript perspective can be easily understood. However, thorough inspection of minor spelling/grammar and text editing mistakes should be performed.

Few minor modifications or explanations are therefore necessary before publication.

Introduction

Line 39,40 and later – I suggest you to be more consistent when stating the fungi species: e.g. F. langsethiae, but Fusarium avenaceum, Fusarium poae, and then again Fusarium langsethiae (Line 45).

Line 69 – When mentioning that there’s no maximum level for HT2+T2, you should state if there is perhaps an indicative level, and also the existing legislative maximum permitted level for DON (µg/kg or mg/kg), and even mention the requirements of the EU legislation.

Line 73 – No need for FHB full name (already mentioned in the Abstract), abbreviation is enough.

Discussion

Line 393 – A year of study by Herrmann et al. (xxxx) should be inserted, also further in text and for other researches/authors (e.g. Tekle, Hautsalo).

Conclusion

Line 601 – fusarium head blight - use abbreviation FHB, or at least capital F in fusarium head blight.

Material and Methods

Line 619 – Please specify if any legislative regulations were followed during sampling or some internal procedures were applied.

Line 641 – Please provide more data regarding the method of analysis, such as sample preparation, etc. Moreover, the stated LOQ of 1 ppb for both DON and HT2+T2 (these two not analysed separately in LC-MS?) sounds very unlikely, could you elaborate?

Lines 638/643 – LOQ, LOD? Abbreviations are to be stated when first time used, please check the whole manuscript and adjust accordingly.

Line 644 – Please provide actual concentrations of DON and HT2+T2 present in the PT sample, and comment the achieved z-score values (<0.35). Brief method validation data could be also added to prove method suitability for mycotoxin determination (both ELISA and LC-MS).

Whole manuscript

A variety of text editing and related mistakes found, some examples stated below, please revise throughout manuscript.

Line 132, 134 – Different fonts; Line 140 – an extra space blank (R2adj =0%, P=0.55); Line 153 and others – 176 μg per kg should be 176 μg/kg; Line 417 - <2,2mm (no space); Line 591 - oat is_; Line 615 – Double space between words; Line 621 - 200g (no space), Line 633 – Instead of ml, you should write mL.

Author Response

Thank you for the constructive feedback you gave on our manuscript. Based on your feedback, we have now corrected the manuscript.

Fungal names:

We understand that the mentioning of fungal names (abbreviated or not) seems inconsistent. The reason why is that we have included the full fungal name (Fusarium graminearum etc) when a fungus is mentioned for the first time and also if the fungal species is mentioned in the beginning of a sentence. Otherwise, the abbreviated names (F. graminearum etc) are used. We have also included the full fungal names in the texts for figures and tables. The reason why there is a blend of abbreviated and non-abbreviated fungal names in line 39- 40 is that some of these fungal species were already mentioned in the text, whereas others were mentioned for the first time.

Legislative and indicative levels:

Thank you for the input. indicative level for HT2+T2, and the existing legislative maximum permitted level for DON are now mentioned in the text. (Text in red in the revised version of the manuscript).

No need for FHB full name when already mentioned in the abstract?

We have now removed the full named of the fungal species and the mycotoxins etc which has already been mentioned in the abstract. However, sometimes journals request that names should be mentioned in full at first mention in main text.

Abbreviations:

Abbreviations are now stated when first time used

References:

The publication ID is now inserted in the text if a study is mentioned followed by et al.

Material and Methods

More data regarding the method of analysis is now included (Text in red in the revised version of the manuscript).

Question from reviewer: ..the stated LOQ of 1 ppb for both DON and HT2+T2 (these two not analysed separately in LC-MS?) sounds very unlikely, could you elaborate?

Our answer: The LC-HRMS method meets very low LOQs do to: 1) using LC-HRMS in the targeted SIM mode i.e. in the orbitrap detecting only the targeted precursor ions, which have higher m/z masses – and better sensitivity - than the smaller product ions usually detected in regular LC-triple quadrupole-MS/MS methods, 2) dissolution of the samples in 25% acetonitrile in water is a pre-requisite for obtaining the low LOQs (using e.g. pure acetonitrile as solvent achieves much poorer LOQs), and 3) using ammonium acetate as a mobile phase, and detecting DON as acetate-adduct and HT-2 and T-2 as NH4-adducts, has (for us) proven clearly superior to gain high sensitivity and low LOQs.

Text editing:

The text is now corrected according to the reviewer’s suggestions
